# A Review of Ambient Air Pollution Exposure Assessment Methods in Determining Childhood Respiratory Health Effects in Children under Five

Sheena Muttoo [1], Prakash M. Jeena [2], Martin Röösli [3,4], Kees de Hoogh [3,4] and Rajen N. Naidoo [1,*]

[1] Discipline of Occupational and Environmental Health, School of Nursing and Public Health, University of KwaZulu-Natal, Room 321, George Campbell Building, Durban 4041, South Africa
[2] Discipline of Paediatrics and Child Health, Nelson R Mandela School of Medicine, University of KwaZulu-Natal, Durban 4041, South Africa
[3] Swiss Tropical and Public Health Institute, 4051 Basel, Switzerland
[4] Faculty of Science, University of Basel, 4001 Basel, Switzerland
[*] Correspondence: naidoon@ukzn.ac.za; Tel.: +27-312604095

**Abstract:** Various epidemiological studies have reported on air pollution exposure-related lung function decline and respiratory health effects in children. Children have increased susceptibility to ambient air pollutants as physiological and structural changes of the lung are still occurring within the first five years of life after birth. This review examines applications in air pollution exposure assessment methods when evaluating lung function and respiratory health concentration–response effects in young children, while considering the effects of critical windows of exposure. We identified 13 studies that used various methods of exposure assessment in assessing respiratory health outcomes (presence of lower respiratory tract infections, respiratory symptoms, wheezing and asthma) in children under five. The methods applied included personal monitoring ($n = 1$), proximity-based methods ($n = 3$), inverse distance weighting ($n = 2$), geographic weighted regression ($n = 1$), dispersion modeling ($n = 1$), satellite-based methods ($n = 2$) and land use regression modeling ($n = 5$). These studies assessed exposure and outcomes at different "windows of susceptibility": antenatally/specific trimesters ($n = 8$), infancy ($n = 5$) and early childhood ($n = 6$). In most studies, the reported measures of air pollutants were noted to be below the prescribed limits, though for some, a cause–effect association was observed. It was also noted that there was very little variation in estimates between time points or trimesters of exposure, likely attributed to limitations in the selected exposure assessment method. Moderate to high correlations between trimesters were reported for most studies.

**Keywords:** environmental exposure assessment; child health; respiratory health

## 1. Introduction

In 2016, the World Health Organization (WHO) reported 543,000 deaths of children under the age of five years, attributed to respiratory tract infections resulting from exposure to air pollution. The burden of impact is particularly high in low- to middle-income settings [1]. Children are notably more vulnerable to air pollution insults as their lungs are still developing at birth [2], with physiological and structural changes still occurring within the first five years of life [3]. Their susceptibility to adverse respiratory health outcomes in early childhood [4] is influenced by factors that occur in utero [5], perinatally [6] and in the early postnatal [7] developmental stages.

Understanding the ambient pollution exposure–lung response relationships perinatally through to early childhood is, therefore, important. While the association between exposure to air pollutants and adverse respiratory outcomes and specifically lung function deficits (as a marker of lung growth) is well documented in older age groups [8–10], this is

less so in very young children. Lung function is considered an important objective marker for respiratory health and a predictor of respiratory morbidity [11]. Children under the age of five, with a still developing lung structure, are particularly at risk for exposure [12,13]. A number of studies have linked air pollution exposure during pregnancy, perinatally and postnatally to lung function decline and respiratory health outcomes [9,14–16]. Recent studies have demonstrated low dose effects of air pollution below the prescribed air quality guidelines and standard levels, to be associated with adverse respiratory outcomes [17]. Specifically, exposure to nitrogen dioxides ($NO_2$) [18] and particulate matter ($PM_{10}$ and $PM_{2.5}$) [19] have been frequently reported on in relation to respiratory health outcomes [20,21], given their association with urban development and as known markers of traffic and industrial pollution.

Understanding the concentration–response effect of an exposure–outcome association for various critical windows of development requires reliable and detailed quantification of the air pollutant of interest. In epidemiological studies, this has been achieved by employing methods of exposure assessment, which vary in their complexity and capability in achieving robust measures of exposure. Air pollution exposure assessment seeks to determine the concentration of pollutants an individual comes into contact with and the duration of exposure, over the period in which such exposure is likely to cause the outcome of interest [4]. Exposures are known to vary in space and time [22], with ambient air pollution dispersion also known to be influenced by seasonal gradients, meteorological and topographical differences. In addition, for the growing lung temporal changes in exposure need to be characterized, from exposure in utero through to birth, neonatal, infancy and early childhood. Thus, the selection of exposure assessment techniques should be informed by these considerations.

Studies have further cited specific periods or "windows" of exposure as having a significant impact on developmental changes, as lung immaturity and physiology of the growing fetus [13,23] and that of very young children predisposes them to increased susceptibility to insults by toxicants [24]. Normal lung development is essential for long-term respiratory health, as significant developmental changes in respiratory physiology are known to occur progressively during the first years of life after birth [13,24–27]; thus, early clinical assessment of lung function is critical in determining the early life risk factors that predispose lung impairment.

This review examines current applications in air pollution exposure assessment methods when evaluating lung function and respiratory health concentration–response effects in children under five, while considering the effects of critical windows of exposure.

## 2. Windows of Susceptibility

The pregnancy period is marked with significant developmental changes that may be affected by exposure to air pollutants. There are a range of respiratory health outcomes reported to be associated with specific windows of exposure to air pollutants $NO_2$ and PM. These are classified as prenatal [5], perinatal [6] and postnatal [7] exposure windows. The adverse changes are structural and functional, with chronic clinical outcomes occurring over time (Figure 1). Damage to the lungs during periods of susceptibility in childhood may result in airway remodeling, which might increase vulnerability to later life insults [2]. Reduced airway caliber and airflow restriction resulting from airway remodeling may manifest as airway hyperresponsiveness, as well as structural changes to the alveoli and supporting parenchyma [3]. Thus, exposure during these periods could impact the development of alveoli and lung growth.

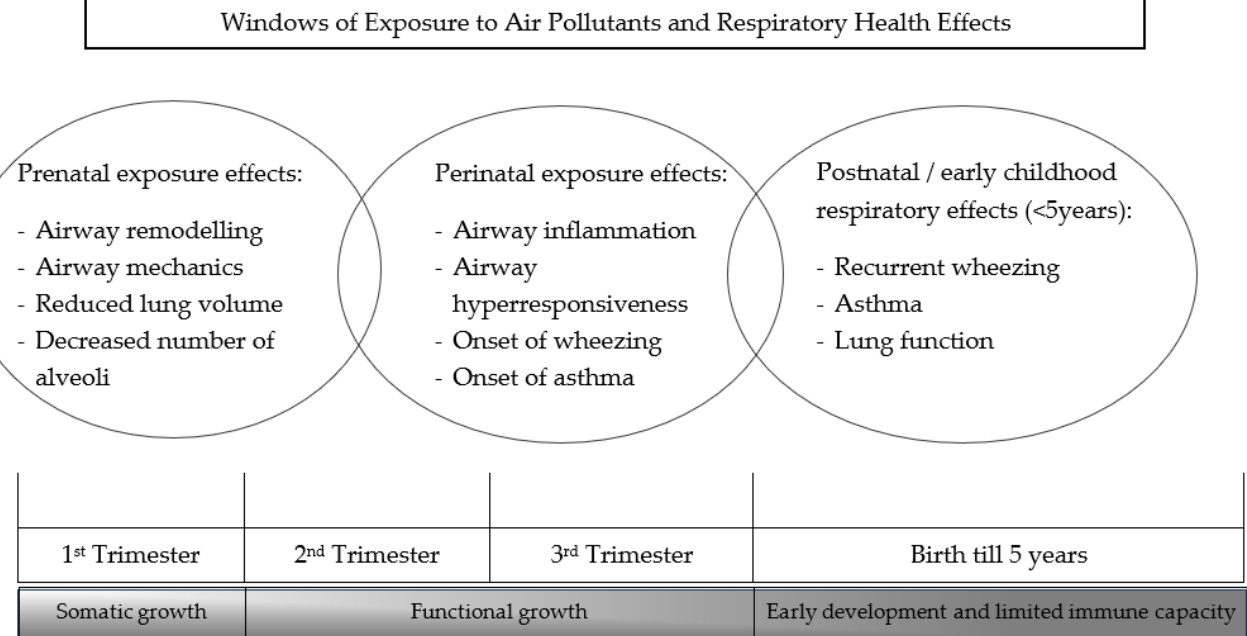

**Figure 1.** Summary of respiratory health effects of air pollution by exposure windows by exposure windows showing prenatal [4,5], perinatal [6,28] and postnatal/early childhood respiratory effects [3,7].

In addition, as the breathing pattern of children differs from that of adults, this may alter the deposition of inhaled pollutants. Children also have a larger surface area per unit of body weight than adults, and under normal breathing, they breathe considerably more air per unit of body weight than adults, thus having a higher breathing rate [2]. Prenatal factors are more likely to affect airway development, while postnatal factors tend to affect airway growth and alveolarization [3]. Diminished airway function identified soon after birth, before any postnatal insult has occurred, may predispose wheezing and diminished lung function in later life. Though studies may not prove causation, they do suggest a dose response effect from specific exposure windows of development that will have varying effects on pre- or postnatal development.

In this review, we provide an overview of current methods in environmental exposure assessment and review the approaches used in studies evaluating air pollution-related lung function decline and respiratory health outcomes in children under five years of age.

### 3. Selection of Studies in this Review

We conducted this review using the "Preferred Reporting Items for Systematic Reviews and Meta-Analyses" (PRISMA) [29] approach (Appendix A) in the selection of articles. Relevant articles were identified through searches on electronic databases including PubMed, Science Direct, Scopus and Google Scholar, as well as reviewing those identified in the reference list of individual articles and other reviews. In our search strategy, we used a combination of search terms, including "outdoor or ambient air pollution", "traffic pollution", "nitrogen dioxide", "particulate matter", "exposure assessment" and "pregnancy exposure", "childhood exposure", "childhood respiratory health" or "childhood lung function". We further applied filters, limiting this review to those articles published in the "English" language, as "observational studies" or "journal articles", among "human" participants with "age" specified as children under five. The search strategy and identification of articles were done without time limits. The identified articles were further assessed to meet our inclusion criteria of presenting (1) validated exposure assessment methods for air pollutants nitrogen dioxide and/or particulate matter; (2) assessing exposure during specific windows including either trimesters of pregnancy, entire pregnancy, infancy or

early childhood; (3) with health endpoints including lung function and respiratory health outcomes (presence of lower respiratory tract infections, respiratory symptoms, wheezing and asthma) assessed in children under five years of age. For studies that referenced their exposure assessment from a separate publication, we further assessed these articles and made reference to them where the exposure assessment for the study was evaluated.

## 4. Results

The initial search identified 5401 articles through the application of selected search terms, and an additional 24 articles were identified through other sources (e.g., bibliographies of other review studies). Furthermore, after the application of relevant filters as described above, 3042 articles were excluded. The remaining 2401 were screened based on their title and abstract to meet our inclusion criteria as described above, resulting in an exclusion of a further 2317. The remaining 84 full text articles were then further assessed, and 71 did not provide adequate details on the exposure assessment. The 13 articles presented in this review met the inclusion criteria for this review (Appendix A).

The selected studies used varying exposure assessment methods. Almost all the studies used routine ambient air quality monitoring networks either as a direct measure of exposure, or for model development and validation ($n = 10$). Only one study used personal monitoring, while others reported the use of proximity-based methods (e.g., distance to nearest emission source) ($n = 3$), inverse distance weighting ($n = 2$), geographic weighted regression ($n = 1$), dispersion modeling ($n = 1$), satellite-based methods ($n = 2$) and land use regression modeling ($n = 5$). These studies assessed exposure and outcomes at different "windows of susceptibility": antenatally/specific trimesters ($n = 8$), infancy ($n = 5$) and early childhood ($n = 6$).

## 5. Personal Monitoring

Only a single study reported measures of personal monitoring. In a study among 336 Polish pregnant women, personal monitors were used to measure exposure to $PM_{2.5}$ over a single 48-h period in the second trimester of pregnancy. A subsample of 80 had single repeated measurements in each trimester. The outcome of interest in this study was lung function [30] of the offspring at the age of five years, while a second report on the same cohort considered wheezing [31] among four year old's.

The subset of repeat personal sampling was intended to determine the representativity of the second trimester sampling for the entire pregnancy. The authors report that within this subset, a consistent trend across the trimesters was present ($PM_{2.5}$ mean (standard deviation) in the second trimester was 42.3 $\mu g/m^3$ (30.8 $\mu g/m^3$) and 38.5 $\mu g/m^3$ (29.9 $\mu g/m^3$) in the third trimester, while the mean difference was not statistically significant (t = 1.015, $p = 0.313$)) [31], strengthening their case of representativity of sampling in the second trimester. However, while the subset sampling provides some confidence in the representativity of exposure, the lack of intra-trimester repeat sampling provides little understanding of variability at this interval. Thus, in measuring the exposure–outcome relationship, the assumption in this study of consistent exposure throughout the pregnancy is responsible for the increased risk, either in adverse lung function or repeated infection. Given that these outcomes are meant to be proxy markers for either abnormal lung growth or development (as opposed to a more acute functional or acute inflammatory response), these once-off exposure measures may be biased.

## 6. Routine Air Quality Monitoring Network Data

In the report of a birth cohort in Bern, Switzerland [32], weekly exposure to $NO_2$ and $PM_{10}$ was assessed from daily mean averages obtained from two local monitoring stations, as well as the development of ten-day lag structures preceding the participant interview date. Because the outcome of interest was acute (respiratory symptoms and infections in children in the first year of life), the use of these short-term exposure measures was used based on the assumption that routine monitoring measurements reflects the temporal

exposure fluctuation in the study region. The 366 infants participating in the study were selected from urban and rural settings. An urban and a rural monitoring station from the national network were used, and exposure of infants was assigned based on the geocoded location in proximation to the two monitoring stations. The use of two monitoring stations to describe exposure among the full cohort of urban and rural resident participants is likely to have resulted in exposure misclassification, given that participant exposure is likely to vary between rural and urban areas. However, the use of the repeated measures design provided a large number of observations ($n$ = 19,106); moreover, the lagged exposure approach is expected to produce reasonable ranking of short-term variability in exposure for evaluating acute exposure–outcome associations.

In describing slightly longer exposure periods (trimesters in pregnancy), a cohort study in Singapore [33] extracted daily 24-h national averages from eight regulatory monitoring stations (from the national network) for the first trimester of each participant. This was used to explore associations between first trimester exposure to $PM_{2.5}$ and increased wheezing episodes in the first two years of life. The use of a national average to describe individual exposure is likely to provide very crude estimates for a three-month period, despite Singapore's geographical size. In addition, spatial variability in estimates was not accounted for. Furthermore, the investigation of a relationship between an antenatal measure of exposure and an acute postnatal outcome implies the hypothesis of an anatomical basis for the outcome. In such instances, control of antenatal factors associated with the acute outcome, including exposure, is critical.

## 7. Proximity and Interpolation Methods

In a study of childhood asthma (at the age of 3–4 years) related to in utero and first year of life exposure among children in British Columbia [34], multiple approaches to exposure characterization were undertaken. The proximity estimates included proximity to roadways (defined as a residential address within 50 m or 150 m of highways and major roads) and proximity to industrial point sources (point sources were assigned an index value based on its pollutant contribution relative to other point sources in the region), with inverse distance weighted (IDW) summation of emissions from other point sources, as a proxy measure of exposure. Effect estimates associated with road proximity failed to reach statistical significance, with confidence intervals including estimates of no effect. However, the IDW models showed statistically significant exposure–outcome associations. In the previously described Swiss BILD cohort [35], among the various methods of exposure characterization, distance of the residential address to the nearest major road of at 6 m width (first class road) and 4 m width (second class road) were used as a proxy for traffic-related exposure. The outcome of interest in this report was lung function in 5-week-old newborns. As with the British Columbia cohort, this particular measure of exposure failed to show any association, despite the suggestion of a trend of increasing risk from the second class to the first class road. There are several reasons why this proximity approach may not be adequate for characterization of exposure: while road distance itself is a crude measure, and road class may improve this, it remains a proxy for vehicle density and vehicle type. These may result in substantial exposure misclassification bias, although part of the error may be of the Berkson type, which does not result in a downward bias but in an increase of the confidence interval [36]. The difference in approaches between studies further show that there is no standardization on how sources may be defined (e.g., major vs minor roads or heavy vs light industry), or at what distance an effect is likely to occur.

Inverse distance weighted methods were used in another study, investigating in utero exposure to air pollutants and incidence of asthma, rhinitis and eczema among 3–6-year-old Chinese children. Daily 24-h average estimates for the duration of the pregnancy, from seven air quality monitoring stations within <5 km radius were interpolated based on proximity of the station to the attending kindergarten school [37]. Monthly means were calculated for the entire pregnancy and individual trimesters. There was no association observed for particulate matter generally, while asthma was associated with $SO_2$ exposure

in the single pollutant models and $NO_2$ was generally associated with all outcomes, but varied across the single and multi-pollutant models. The absence of effect with particulates and $SO_2$ could be interpreted as a true absence of effect, or that the IDW approach from station to kindergarten did not sufficiently represent exposure. The primary difference between the British Columbia and China studies is that the former determined IDW estimates based on proximity to industrial point sources and roadways, while the latter used proximity to air quality monitoring stations to assign exposure, and this may have resulted in misclassification of exposure.

## 8. Land Use Regression Models

Land use regression has been used as a predictive modeling tool, which relies on monitored data at selected sites and geographic predictors as input data into a stochastic model that is able to predict concentrations at unmonitored locations within a study domain. We identified five reports (two of which are from the same cohort), in addition to the British Columbia, Canada study described above, that reported on the use of land use regression (LUR) modeling [14,16,34,38–40]. Among these, three studies [34,39,40] used air quality monitoring networks (AQMN) to develop their models with the number of monitors used ranging from 7 to 78. The outcomes of interest in these studies were asthma. The remaining studies [14,16,38], which focused on lower respiratory tract infections, wheezing and lung function, used passive sampling measures for model development. Apart from the Spanish and Norwegian studies, all studies used a combination of approaches to supplement the LUR approach, thus accounting for spatio-temporal variability. These included the use of satellite-based estimates as observed in the Ontario, Canada [40] and the Boston, USA [39] studies. Inverse distance weighting was additionally used in the latter two studies as well as the British Columbia, Canada study [38]. The combination of methods improves upon the spatial and temporal variability in exposure estimates. The studies that did not use supplementary approaches for more temporally refined data, were shown to have exposure estimates that were highly correlated (between-subjects and between-trimesters). This is likely attributed to limited variability in AQMN data, while studies with fewer sites are subject to increased exposure misclassification due to the sparse distribution of these sites. Furthermore, inadequate representation of the different windows of exposure are unlikely to provide accurate exposure–outcome effect estimates. Studies that used LUR on its own found greater difficulty in disentangling trimester-specific effects given the limited temporal variability in exposure estimates. The success of LUR models is largely dependent on the accuracy of input data.

## 9. Dispersion Models

In our review, we identified only one study to use dispersion modeling [41]. In this study of pollutant-related wheezing in Dutch preschoolers, ambient pollution exposure in each year of the first three years of life was determined. Measures of traffic intensity and emissions, meteorology patterns, shipping, industry and household data were included as input data in the model, which further adjusted for background concentrations from three regulatory air quality monitors with incorporation of the Dutch standard methods [42] (inclusion of intra-urban road traffic, traffic on highways, and industrial and other point sources), thus accounting for temporal changes in air pollutant exposure. The model was then validated by comparison between predicted annual average $PM_{10}$ and $NO_2$ concentrations, and measured data from the available monitoring stations. This study found no association of effect with the preceding year's average air pollutant exposure, but a statistically increased risk was observed with the preceding monthly averages for $NO_2$. The lack of a consistent exposure-response effect probably reflects the challenge in characterizing the appropriate exposure metric for the outcome of interest: wheezing in early infancy is probably the clinical manifestation of a long-term structural damage of the infant airway, and long-term exposure is probably a better exposure metric to understand this relationship. However, wheezing is also an acute outcome as a result of a recent insult

(e.g., infection, recent air pollutant exposure, etc.), and a shorter (recent)-term exposure is more appropriate. In this study, it is likely that the preceding yearly average failed to capture the exposure required for a structural impairment, but better described the acute outcome.

## 10. Remote Sensing

In a pregnancy cohort conducted in Boston, USA [39], estimates of $PM_{2.5}$ exposure was determined by spatio-temporal modeling incorporating MODIS (moderate resolution imaging spectrodiameter) satellite-derived AOD (aerosol optical depth) measurements. AOD data account for temporal variability in air pollutant emissions. The AOD-$PM_{2.5}$ data were calibrated for daily estimates using grid cells and AOD values by mixed modeling with random slopes. The objective was to identify sensitive antenatal exposure windows for the development of asthma by the age of six years in the cohort. The predicted satellite data were further combined with LUR derived data, providing modeled estimates at $10 \times 10$ km$^2$ spatial resolution, for pregnancy exposure to $PM_{2.5}$. In a retrospective study in Canada [40], satellite estimates were derived at a $1 \times 1$ km$^2$ spatial resolution, for $PM_{2.5}$ used in combination with a chemical transport model, with further adjustment by geographic weighted regression. The resulting time-varying (during pregnancy, the first year of life and in childhood) exposure estimates were then linked to childhood asthma [40]. The strengths of this exposure characterization approach were the use of multiple sources and methods of validation achieving fine temporal granularity, with the LUR layered approach providing the required spatial estimates. AOD reflects air pollution in the atmosphere, and thus, calibration is needed to obtain ground level air quality data, which may introduce some exposure assessment errors. An increased antenatal pollutant-related risk was found almost across the entire antenatal period, but this was statistically significant in the 16–25 weeks of pregnancy. The relatively narrow confidence intervals (ranging from approximately 0.8–1.3), although including the null effect, suggests robust effect estimates.

## 11. Discussion

Exposure assessment in environmental epidemiology facilitates the investigation of a cause–effect relation between an environmental toxicant and an adverse health outcome. The relevance of its application in pregnancy or childhood studies is based on evidence suggesting early programming effects during the prenatal [4,5], perinatal [6,28] and early postnatal [3,7] windows of development, which may result in structural and physiological changes in the airways and lungs, with implications for long term respiratory health. The sensitivity of specific windows of exposure (trimesters of pregnancy or early childhood) and associated respiratory health outcomes suggests that time integration is critical in exposure assessment in accounting for inter- and intra-subject variability. Individual personal monitoring lacks logistical feasibility for large scale studies; thus, modeling methods are necessary to simulate and predict air pollutant exposure based on known characteristics of the surrounding environment. Despite significant advancements in this field, a multitude of challenges still exist in addressing individual level exposure estimation, as opposed to aggregate population level exposure, and accounting for temporal variability in exposure estimates.

The time-point of interest and duration of exposure varies across studies, ranging from specific trimesters or entire pregnancy, shortly after birth or early childhood, depending either on the objectives of the study, or the availability of either exposure or outcome data [43]. Assessing exposure during pregnancy is further influenced by fetal susceptibility and other maternal and biological risk factors [3]. Exposure assessment has a significant role to play in identifying dose-response effects; thus, highly resolved fine scale spatio-temporal estimates are required. Acute effects such as lower respiratory tract infections, respiratory symptoms or wheezing can be assessed in early infancy and may be linked to in utero exposure or recent exposure events. Studies assessing long-term health outcomes,

such as asthma, may look at developmental changes over time, thus having highly resolved time-relevant exposure data available, and may help identify when such changes are occurring [4].

Because the lung begins growth and development antenatally, and continues to develop postnatally in early infancy, the effects of external insults such as air pollution may have an impact at specific time points, either once or at multiple points, or throughout the developmental period [44].This raises the complexity for determining exposure–outcome relationships for lung health specifically or other organ health generally in this perinatal period. Arbitrary choices of cross-sectional exposure metrics, repeated measures of exposure at selected timepoints or even assumptions of averaging exposure over extended periods are likely to result in exposure misclassification to some extent. As shown in Table 1, at least seven of the studies reviewed reported moderate to high correlation of measured observations between trimesters. Air pollutants are likely to covary, given that they are emitted from the same sources or produced by similar atmospheric chemistry or meteorologic processes; however, their chemical and physical properties are likely to yield differing impacts on the severity of health outcome assessed. For example, exposure to particulates are associated with lung inflammation and mucous secretion by acting on airway epithelial cells and alveolar macrophages with the potential of leading to airway remodeling [19]. Nitrogen dioxide exposure is associated with increased incidence in lower respiratory tract infections in children and increased airway responsiveness in asthmatics [45,46]. Correlation between pollutants make it difficult to assess individual or combined health effects, as estimates may become unstable when adjusting for multiple pollutant effects in regression analysis.

Research has suggested that given that humans are simultaneously exposed to a complex mixture of air pollutants, "multi-pollutant" approaches should be considered [47]. A central aspect of multi-pollutant approaches is to model complex air pollution mixture effects more explicitly to gain better insight into the features that define the toxicity of an air pollution mix [48]. This approach may characterize more fully the complexity of the exposure and the health outcomes, with the potential to identify the most harmful emission sources. While this has yet to be fully explored, new approaches should modify the current methods of specifying air pollutant concentrations (or exposures) in statistical models to estimate health effects.

**Table 1.** Detailed Summary of Studies Reviewed.

| Author | Study Area | Study Design/ Assessment Age | Health Outcome & Effect Estimates | Pollutant | Pollutant Data/Mean (SD)/Median (IQR) | Exposure Estimation Method | Temporal Adjustment | Additional Information |
|---|---|---|---|---|---|---|---|---|
| Soh et al., 2018 [33] | Singapore | Longitudinal birth cohort—Growing Up in Singapore towards health Outcomes (GUSTO) Assessment age—2 years | Wheezing episodes | $PM_{2.5}$ | $PM_{2.5}$ (µg/m³): 17.92 (1.31) * 18.21 (2.97) ** :18.24 (2.68) *** 17.17 (2.39) **** | (a) National AQMN (*n* = 8): daily 24-h average (2009–2013) | Trimesters of pregnancy. | $PM_{2.5}$ between trimesters was moderately correlated and strongly correlated within a trimester; thus, multi-trimesters were adjusted for in the models. |
| Lavigne et al., 2017 [40] | Ontario, Canada | Retrospective cohort Assessment age—birth to 6 years | Asthma | $NO_2$ $PM_{2.5}$ | $NO_2$ (ppb) 13.2 (7.8) * 13.2 (7.8) ** 13.2 (7.8) *** 13.1 (7.8) **** 13.1 (7.8) ***** 13.0 (7.8) ****** $PM_{2.5}$ (µg/m³): 7.3 (3.0) * 7.3 (3.0) ** 7.3 (3.0) *** 7.3 (3.0) **** 7.3 (3.0) ***** 7.3 (3.0) ****** | (a) Satellite AOD estimates at 1x1 km resolution (2006–2012) (b) GWR were used to determine $PM_{2.5}$ exposure estimates (c) National LUR was developed using AQMN data (*n* = 46), satellite estimates (2005–2011) and spatio-temporal characteristics (road length, industrial land use, mean summer rainfall) to determine $NO_2$ exposure estimates LUR Adjusted R2 = 0.73 (for 2006) (d) IDW—applied to zip codes within 25 km of the AQMN to create a scaling surface | Trimesters of pregnancy, first year of life; cumulative childhood. A scaling factor used by calculating ratio of monthly mean $NO_2$ concentrations per monitor used to adjust the LUR estimate by trimesters of pregnancy. | $PM_{2.5}$ was moderately correlated with $NO_2$ during the entire pregnancy period. Moderate correlations were observed between trimester-specific periods and exposures after birth to $PM_{2.5}$. |
| Madsen et al., 2016 [16] | Norway | Prospective population-based pregnancy cohort—Norwegian Mother and Child Cohort study (MOBA) Assessment age: birth to 18 months | Lower respiratory tract infection and wheezing | $NO_2$ | $NO_2$ (µg/m³): 13.6 (6.9) * 13.7 (7.4) ** 13.8 (7.5) *** 13.6 (7.3) **** | (a) LUR—$NO_2$ measured by passive samplers (Oslo *n* = 14; Arkerhus *n* = 36; Bergen/Hordaland *n* = 46); three sampling campaigns of two-weeks each, over the duration of a year (2010–2011) during summer, winter and an intermediate season. LUR Adjusted R² Oslo = 0.65 LUR Adjusted R² Arkerhus = 0.55 LUR Adjusted R² Bergen/ Hordaland = 0.85 | AQMN data (2000–2012) were used for the ratio method of back-extrapolation during the pregnancy period—the LUR-modeled yearly estimate multiplied by the ratio between daily $NO_2$, AQMN data and an annual average for the year of LUR measurement campaign. Daily $NO_2$ exposure estimates were averaged separately for the first, second and third trimester, as well as the entire pregnancy [49]. | Exposures by trimester and entire pregnancy exposure were highly correlated. Thus, average $NO_2$ exposure during entire pregnancy was used as the exposure estimate in the analyses. |
| Deng et al., 2016 [37] | Hunan Province, south-central China | Survey study Assessment age: 3–6 years | Asthma | $NO_2$ $PM_{10}$ | $NO_2$ (µg/m³): 46.0 (8.0) * 45.0 (11.0) ** 46.0 (11.0) *** 46.0 (10.0) **** $PM_{10}$ (µg/m³): 110.0 (11.0) * 113.0 (16.0) ** 110.0 (15.0) *** 108.0 (18.0) **** | (a) AMQN (*n* = 7)—daily averages (2005–2008). Spatial resolution—1909 km² (b) IDW used to establish individual exposure estimates | Average of the monthly mean concentrations of AP was calculated for trimesters of pregnancy and entire pregnancy. | The pollutants during each trimester were weakly or moderately correlated with each other. Each pollutant was also weakly or moderately correlated between different trimesters. Multi-pollutant models were explored. |

**Table 1.** *Cont.*

| Author | Study Area | Study Design/ Assessment Age | Health Outcome & Effect Estimates | Pollutant | Pollutant Data/Mean (SD)/Median (IQR) | Exposure Estimation Method | Temporal Adjustment | Additional Information |
|---|---|---|---|---|---|---|---|---|
| Hsu et al., 2015 [39] | Boston, USA | Pregnancy cohort—Asthma Coalition on Community, Environment and Social Stress (ACCESS) Assessment age—6 years | Asthma | $PM_{2.5}$ | $PM_{2.5}$ (µg/m³), median (IQR) 11.2 (10.2–11.8) * | (a) MODIS satellite-derived AOD measurements at $10 \times 10$ km resolution (b) LUR derived using AMQN data ($n = 78$) meteorological variables combined with AOD-$PM_{2.5}$ measurement was calibrated daily. | Entire pregnancy and distributed lag windows over 6 years | |
| Morales et al., 2014 [14] | Sabadell and Gipuzkoa, Spain | Cohort study—INMA (Environment and Childhood) Study Assessment age: 4–5 years | Lung function | $NO_2$ | $NO_2$ (µg/m³), median (IQR): 25.50 (17.40–31.66) * 24.30 (16.76–33.48) ** 24.23 (16.96–25.63) *** 23.87 (16.88–33.26) **** 27.87 (19.84–33.59) ***** | (a) LUR ambient $NO_2$ measured by passive samplers ($n = 57$) (2005–2006) [50] during four sampling campaigns of one week each LUR Adjusted $R^2$ Sabadell = 0.75 LUR Adjusted $R^2$ Gipuzkoa = 0.51 | Adjustment using a ratio of daily $NO_2$ levels from AMQN to establish estimates for the entire pregnancy, the trimesters of pregnancy and the first year of life | $NO_2$ levels were moderately tohighly correlated between trimesters of pregnancy, and highly correlated between the entire prenatal period and the first year of life. |
| Stern et al., 2013 [32] | Bern, Switzerland | Prospective birth cohort—Bern Infant Lung Development Study (BILD) Assessment age: 1 year | Respiratory symptoms | $PM_{10}$ $NO_2$ | $PM_{10}$ (µg/m³): Weekly average rural: 19.9 (10) Weekly average urban: 32.6 (13) $NO_2$ (µg/m³): Weekly average rural: 15.2 (7) Weekly average urban: 48.2 (9) Pregnancy Exposure to $PM_{10}$ (µg/m³): Urban: 34.2 (4.2) * Rural: 20.8 (2.5) * | (a) Swiss National Air Pollution Monitoring Network—daily mean hourly data for $PM_{10}$ and $NO_2$ (2004–2006) Proximity measures—distance to nearest major road of 4 to 6 m width | Lag windows of 1–10 days established during the first year of life Lag structures of 1 to 10 days preceding interview were constructed with shifting windows of weekly mean AP by 1–10 days | |
| Aguilera et al., 2013 [38] | Spain | Birth cohort Assessment age: 12–18 months | Lower respiratory tract infections and wheezing | $NO_2$ | $NO_2$ (µg/m³), median Asturias—21.0 * & 22.0 ***** Gipuzkoa—18.0 * & 19.0 ***** Sabadell—30.0 * & 32.0 ***** Valencia—38.0 * & 38.0 ***** | (a) LUR model developed using ambient $NO_2$ measured by passive sampling during four sampling campaigns of one week each. Asturias ($n = 67$) + 4 AQMN Gipuzkoa ($n = 86$) + 2 AQMN Sabadell ($n = 57$) + 1 AQMN Valencia ($n = 93$) + 7 AQMN LUR model $R^2$ Asturias = 0.52 LUR model $R^2$ Gipuzkoa = 0.52LUR model $R^2$ Sabadell = 0.75 LUR model $R^2$ Valencia = 0.73 | Exposure estimate derived by multiplying LUR estimate by the ratio between average measured concentration at AQMN over the pregnancy period to establish trimester-specific and entire pregnancy estimates. | Levels of each pollutant were moderately to highly correlated between trimesters of pregnancy and highly correlated between the entire prenatal period and the first year of life. |
| Sonnenschein-van der Voort, 2012 [41] | Rotterdam, the Netherlands | Prospective cohort—Generation R study Assessment age: 1–3 years | Wheezing | $NO_2$ $PM_{10}$ | $PM_{10}$ (µg/m³): 28.86 (2.11) ^ 28.27 (1.57) ^^ 27.92 (1.67) ^^^ $NO_2$ (µg/m³): 38.66 (4.20) ^ 37.46 (4.17) ^^ 36.22 (4.28) ^^^ | (a) AQMN data ($n = 3$), (taking into account wind conditions and fixed temporal patterns from sources) (b) Dispersion modeling [51]. | Average annual levels per year over 1–3 years. | |

**Table 1.** *Cont.*

| Author | Study Area | Study Design/ Assessment Age | Health Outcome & Effect Estimates | Pollutant | Pollutant Data/Mean (SD)/Median (IQR) | Exposure Estimation Method | Temporal Adjustment | Additional Information |
|---|---|---|---|---|---|---|---|---|
| Jedrychowski et al., 2010 [31] | Krakow, Poland | Birth cohort Assessment age: 4.5 years | Wheezing | $PM_{2.5}$ | $PM_{2.5}$ ($\mu g/m^3$), median (IQR)35.4 (10.3–294.9) | (a) Personal Environmental Monitoring Samplers over a 48-h period (second trimester ($n = 369$); third trimester ($n = 85$)) | 48-h measurement extrapolated over specific trimesters (second and third) | |
| Jedrychowski et al., 2010 [30] | Krakow, Poland | Birth cohort Assessment age: 5 years | Lung function | $PM_{2.5}$ | $PM_{2.5}$ ($\mu g/m^3$), median (IQR)32.4 (30.1) | (a) Personal Environmental Monitoring Samplers over a 48-h period (second trimester ($n = 176$)) | 48-h measurement extrapolated over specific trimesters (second and third) | |
| Clark et al., 2010 [34] | British Columbia, Canada | Nested case-controlAssessment age: 3–4 years | Asthma | $NO_2$, $PM_{10}$ $PM_{2.5}$ | *Controls* $NO_2$ ($\mu g/m^3$) LUR—31.68 (8.64) * & 29.86 (8.85) ***** IDW—30.74 (8.90) * & 29.86 (8.85) ***** $PM_{10}$ ($\mu g/m^3$) IDW—11.94 (1.35) * & 12.37 (1.00) ***** $PM_{2.5}$ ($\mu g/m^3$) LUR—4.67 (2.47) * & 4.50 (2.45) ***** IDW—4.74 (1.19) * & 5.62 (0.61) ***** *Asthma Cases* $NO_2$ ($\mu g/m^3$) LUR—31.73 (8.42) * & 30.68 (9.06) ***** IDW—31.37 (9.20) * & 30.68 (9.06) ***** $PM_{10}$ ($\mu g/m^3$) IDW—12.03 (1.30) * & 12.42 (1.00) ***** $PM_{2.5}$ ($\mu g/m^3$) LUR—4.78 (2.46) * & 4.59 (2.40) ***** IDW—4.71 (1.20) * & 5.62 (0.61) ***** | (a) LUR models derived using AMQN—24-h averages [$NO_2$ ($n = 14$); PM10 ($n = 19$); $PM_{2.5}$ ($n = 7$)]—road density, population density, elevation and type of land use were used to develop high-resolution (10 m) maps (b) IDW-summation of emissions from point sources within 10 km (c) Proximity measures—distance to roadways and industrial point sources within 50 m or 150 m of highways and major roads | Daily average over entire pregnancy and in the first year of life. | Pregnancy and first-year exposures were moderately to highly correlated, some of which could be examined together in mutually adjusted models. Multi-pollutant methods could not be explored due to correlation. |
| Latzin et al., 2009 [35] | Bern, Switzerland | Prospective birth cohort—BILD Assessment age: 5 weeks | Lung function | $NO_2$ $PM_{10}$ | $PM_{10}$ ($\mu g/m^3$), median (IQR) 221. (20.2–23.8) * 20.0 (16.6–23.4) ****** $NO_2$ ($\mu g/m^3$), median (IQR) 15.8 (14.7–17.0) * 15.1 (10.9–19.7) ****** | (a) AQMN in Payern (part of the Swiss National Air Pollution Monitoring Network)—daily mean hourly data for $PM_{10}$ and $NO_2$ (2004–2006). (b) Proximity methods—distance to nearest major road of 4 to 6 m width | Entire pregnancy, trimesters of pregnancy and birth until test date (postnatal). Mean daily levels of AP were established over the estimation period. | |

* entire pregnancy (prenatal/in utero); ** first trimester; *** second trimester; **** third trimester; ***** first year of life, ****** cumulative/postnatal. ^ one year; ^^ two year; ^^^ three year.

## 12. Conclusions

The usefulness of spatio-temporal modeling methods ultimately depends on the epidemiological study design (e.g., case crossover vs time series vs. cohort), the health outcome of interest including acute (e.g., respiratory symptoms or infections and lung function) and chronic (wheezing and asthma) outcomes, the pollutant of interest and their specific spatial and temporal patterns. Robust measures of exposure are critical in epidemiological studies of exposure–health outcomes, and methods that incorporate both spatial and temporal gradients would likely yield better results than simplistic approaches (interpolation, regulatory monitoring data and proximity-based measures), with a further benefit of reducing prediction error or exposure misclassification. Future studies may likely benefit from considering multi-pollutant models to characterize complex pollutant interactions and account for spatio-temporal variability in the emissions of complex mixtures.

**Author Contributions:** Conceptualization, S.M., P.M.J., M.R., K.d.H. and R.N.N.; Methodology, S.M., P.M.J., M.R., K.d.H. and R.N.N.; writing—original draft preparation, S.M.; writing—review and editing, supervision, P.M.J., M.R., K.d.H. and R.N.N.; funding acquisition, R.N.N. All authors have read and agreed to the published version of the manuscript.

**Funding:** This work is supported by the Mother and Child in the Environment (MACE) project funders, the National Research Foundation (Grant 90550), the South African Medical Research Council (SAMRC), AstraZeneca Trust and the University of KwaZulu-Natal's Flagship funding. The support of the Swiss-African Research Cooperation (SARECO) for a fellowship award (to S. Muttoo) and the South African Medical Research Council (SAMRC) researcher development grant (to S. Muttoo) is acknowledged.

**Institutional Review Board Statement:** Not applicable.

**Informed Consent Statement:** Not applicable.

**Data Availability Statement:** Not applicable.

**Acknowledgments:** We acknowledge the extensive support of Professor Bert Brunekreef and the Institute of Risk Assessment Sciences of Utrecht University in the Netherlands for their capacity building contributions and support of exposure assessment within the MACE cohort.

**Conflicts of Interest:** The authors have no conflict of interest to declare.

## Abbreviations

| | |
|---|---|
| AP | Air pollutant |
| AOD | Aerosol Optical Depth |
| AQMN | Air Quality Monitoring Network |
| DM | Dispersion Modeling |
| GWR | Geographic Weighted Regression |
| IDW | Inverse Distance Weighting |
| IQR | Interquartile Range |
| LUR | Land Use Regression |
| MODIS | Moderate Resolution Imaging Spectroradiometer |
| $NO_2$ | Nitrogen Dioxide |
| $PM_{10}$ | Particulate Matter <10 microns |
| $PM_{2.5}$ | Particulate Matter <2.5 microns |
| SD | Standard Deviation |

**Appendix A**

**Figure A1.** Selection of studies for review based on the Prisma strategy.

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
