# Peer review of "A Review of Ambient Air Pollution Exposure Assessment Methods in Determining Childhood Respiratory Health Effects in Children under Five"

_environments, doi:10.3390/environments9080107_

Round 1

Reviewer 1 Report

As a reviewer I have the following remarks. I am using “you” as Dear Authors.

  1. “Quantification of exposure using robust exposure assessment methods is an important component to understanding the dose-response effects of air pollutant exposure on lung function and respiratory health outcomes.” – the sentence very complex, and for me lack of the word “concentration”.
  2. “demonstrated low dose effects” – we don’t know dose, rather concertation.
  3. “Specifically, exposure to nitrogen dioxides…” – I think also ozone.
  4. “while considering the time varying effects of critical windows of exposure.” – for me it’s not clear, time is related to exposure or effect – develop health condition.
  5. On searching, why not ozone and urban air pollution?
  6. Line 47 – no any fixed monitors measurements?
  7. Line 60 - mean (SD), better to say “mean (standard deviation)” as later SD is not used. At least in the first use should be full spelled. Also cubic meters (m^3, now it is m3).
  8. Line 144: “the use of LUR model” should be the use of Land Use regression (LUR) model.
  9. Line 154: British Columbia, Canada, study,,,
  10. Table 1: 25.50 (17.40 – 31.66)* this interval is reported as IQR?, IQR is just one value (no interval). The IQR is the difference between Q3 and Q1.
  11. Lag windows of 1-10 – what’s this?
  12. PM (μg/m3): - which pm?. Table 1.
  13. I think in some places/context should be used the term “concentration”. In reality we don’t know the dose.

Thank you

Reviewer 2 Report

I think the presented study is very useful because it shows us different methods of air pollution exposure assessment used in different studies.

I have few comments regarding the method used to select the studies in this review.

-The authors limited the pollutant to particles and nitrogen dioxide. Ozone is also a pollutant linked to respiratory problems. Why wasn't it included? Authors may add an explanation. I'm not suggesting doing any extra work, only explaining the restrictions.

- Along the same line, some indoor pollutants, such as mold and formaldehyde, are also linked to respiratory problems. I understand that these pollutants are difficult to measure, but they must be mentioned as important causes.

Line 49-50 – “…early infancy (n=5) and early childhood (n=6).” – Please explain the difference between the two groups.

Reviewer 3 Report

Dear Authors,

Please consider the following comments and suggestions.

-Although the focus of the review is on the air pollution exposure assessment methods, a brief description of the main results of the cited works is sometimes missing in the text and should be added;

-DOI must be added in all references;

-In section 4.Results, line 40: Table 1 refers 13 papers (not 17). Please complete Table 1 with the 17 mentioned references;

-Line 82: "The use of two monitoring stations to describe exposure among the full cohort of urban and rural resident participants is likely to have resulted in exposure assessment errors." Please explain;

-Line 86: typo (remove ",");

-Line 136: "The primary difference between the British Columbia and China studies is the former determined IDW estimates based on proximity to industrial point sources and roadways"...

-Line 173: typo (remove ".");

-The Conclusions section should not include references. References 50 and 51 should be included in an earlier stage of the paper.

-Line 5 of Conclusions: typo ("." should be ",")
